# Potential for Protein Kinase Pharmacological Regulation in *Flaviviridae* Infections

**DOI:** 10.3390/ijms21249524

**Published:** 2020-12-15

**Authors:** Ana-Belén Blázquez, Juan-Carlos Saiz

**Affiliations:** Department of Biotechnology, Instituto Nacional de Investigación y Tecnología Agraria y Alimentaria (INIA), 28040 Madrid, Spain; jcsaiz@inia.es

**Keywords:** protein kinases, phosphorylation, antivirals, flaviviruses, hepatitis C virus

## Abstract

Protein kinases (PKs) are enzymes that catalyze the transfer of the terminal phosphate group from ATP to a protein acceptor, mainly to serine, threonine, and tyrosine residues. PK catalyzed phosphorylation is critical to the regulation of cellular signaling pathways that affect crucial cell processes, such as growth, differentiation, and metabolism. PKs represent attractive targets for drugs against a wide spectrum of diseases, including viral infections. Two different approaches are being applied in the search for antivirals: compounds directed against viral targets (direct-acting antivirals, DAAs), or against cellular components essential for the viral life cycle (host-directed antivirals, HDAs). One of the main drawbacks of DAAs is the rapid emergence of drug-resistant viruses. In contrast, HDAs present a higher barrier to resistance development. This work reviews the use of chemicals that target cellular PKs as HDAs against virus of the *Flaviviridae* family (*Flavivirus* and *Hepacivirus*), thus being potentially valuable therapeutic targets in the control of these pathogens.

## 1. Introduction

Kinases are a group of enzymes that catalyze the transfer of the terminal γ-phosphate from ATP to the hydroxyl group of an acceptor substrate, thus participating in a huge variety of cellular processes, such as proliferation, apoptosis, metabolism, transcription, or antibiotic resistance, among others. These phosphorylation reactions can be reversed by the corresponding phosphatases. These important discoveries concerning reversible protein phosphorylation as a biological regulatory mechanism were recognized by the award of the Nobel Prize in Physiology or Medicine in 1992 to Edmond H. Fischer and Edwin G. Krebs [1]. Even though all kinases catalyze the same phosphoryl transfer reaction, there is a wide diversity in their structures and substrates [2], which include proteins, lipids, carbohydrates, amino acids, vitamins, and cofactors. According to their structure and sequence, kinases have been classified into 30 families [3], of which the protein kinase (PK) family is the largest comprising one of the most abundant protein families in mammalian genomes [4].

PKs catalyze protein phosphorylation, mainly of serine, threonine, and tyrosine residues, and play a critical role in cellular signaling pathways that affect crucial cell processes, such as growth, differentiation, and metabolism [5]. Phosphorylated proteins can initiate a downstream cascade of reactions, resulting in a vast range of responses including activation or inhibition of enzyme activities [6], changes in biological activity, as well as facilitating or perturbing movement between subcellular compartments, and initiating or interrupting protein–protein interactions [7].

PK activity was first observed in 1954 in an enzyme that catalyzed casein phosphorylation [8]. The first evidence that one PK can activate another was the observation that cAMP-dependent protein kinase A (PKA) activates phosphorylase kinase [9]. The same kinase, PKA, was also described as the first example of enzyme inhibition by phosphorylation, which inhibits glycogen synthase [10]. Since then, characterization of PKs has been widely addressed. In 2002, Manning et al. [11] published the PK complement of the human genome, the so-called kinome, classifying it in 518 members, with most PKs (478) belonging to a single superfamily containing a eukaryotic PK (ePK) catalytic domain, while the other 40 PK genes were reported to belong to a few atypical families (aPK) with proteins showing biochemical kinase activity, but with no sequence similarity to the ePK domain. According to sequence comparison of their catalytic domains, the human kinome was classified into nine different phylogenetic groups for ePKs (Table 1) and four groups for aPKs (Table 2).

All PKs contain a conserved catalytic core comprised of two lobes (N and C) located, respectively in the N- and C-terminal position of this domain. They may also contain additional, family-specific domains, which can be N- and/or C- terminal to the kinase domain. The N-lobe binds and orientates ATP, and the α-helical C-lobe is essential for substrate binding and phosphoryl transfer initiation [12]. An active-site cleft containing the ATP-binding site is placed between the two lobes [13]. Although the catalytic core can vary in terms of length and sequence among the different PKs, the residues involved in the catalysis are highly conserved [12]. These differences in the catalytic core influence the way in which substrates interact with kinase, thus determining the specificity of kinase-substrate interactions [14]. PKs can exist in an active or an inactive state, and, in many cases, phosphorylation is required for full enzymatic activity. The mechanisms of activation and deactivation are diverse and kinase specific [15,16]. In the active state, all kinases show a similar conformation, however, several differences are observed between inactive states. Both states have been targeted to produce potent and selective chemicals to modulate PKs activity. It has been observed that, overall, compounds directed to the inactive conformation have high specificity, while those acting on the active conformation are less specific and generally target several PKs [6].

Due to the fact that PKs are essential components in cellular processes, they are being thoroughly studied and characterized, since, among other activities, they represent attractive targets for drugs against a wide spectrum of diseases, ranging from cancer to cardiovascular problems, diabetes, or immune disorders [5]. Around one-third of current research in drug discovery is directed against the PK superfamily [17]. Kinases are also a key target for therapies against microbial and viral diseases [18,19]. In this sense, viral infections rely on virus–host interactions, and it has been described that host cell kinases play crucial roles in every step of the viral life cycle in a wide range of viral species, including members of the *Flaviviridae* family (flaviviruses and hepaciviruses) [19,20,21,22]. In fact, viruses control a vast number of host kinases at different steps along their life cycle [23].

The *Flaviviridae* is a family of enveloped positive sense single-stranded RNA viruses that comprise more than 60 species grouped into four genera (Table 3).

The genomes of the members of this family are constituted by a single open reading frame (ORF) flanked by 5′ and 3′ untranslated regions (UTRs, Figure 1) [24]. The ORF is translated into a single polyprotein that forms specific secondary structures required for genome translation, virion assembly, cell receptor binding and entry, polyprotein processing, and viral replication.

The 5′ and 3′ UTRs are implicated in viral replication and pathogenesis [25]. The capsid (C) protein plays an essential role in viral assembly and replication [26]. The membrane (M) and polypeptide 7 (p7) proteins participate in viral assembly and release of infectious virions [27,28]. The envelope (E) protein, the most immunogenic of the viral proteome, is involved in receptor binding, viral entry, and membrane fusion [27,29]. The non-structural (NS) proteins are implicated in viral RNA replication, virulence, immunomodulation, viral assembly and modulation of cellular processes [27,29,30,31].

The *Flavivirus* genus comprises more than 50 members [32], some of which are human pathogens, causing life-threatening diseases, such as yellow fever, dengue, Japanese encephalitis, West Nile encephalitis, and Zika disease [33]. Flaviviruses are arboviruses (arthropod-borne viruses) mainly transmitted by mosquitoes, and, not surprisingly, due to global climate warming and increasing travelling and trade, their geographic distribution is growing. Many flaviviruses are zoonotic, such as West Nile virus (WNV), with birds as the main natural host [34,35] or Japanese encephalitis virus (JEV), with a cycle involving aquatic birds and pigs as amplifying hosts [36]. Yellow fever virus (YFV) has a sylvatic cycle, which serves to maintain the virus in wild reservoirs between outbreaks in humans [37]. Flaviviruses cause globally relevant epidemics in humans, infecting up to 400 million people annually [33]. Dengue virus (DENV), classified in four serotypes and currently endemic in more than 100 countries [38], can cause a wide spectrum of disease manifestations ranging from a subclinical self-limited infection or a mild febrile illness termed dengue fever, to a life-threatening dengue hemorrhagic fever and dengue shock syndrome, especially after secondary infections with an heterologous serotype [39]. Zika virus (ZIKV) spread throughout the American continent in 2015 causing considerable worldwide social and medical alarm due to its association with congenital disorders [29], such as microcephaly in newborns, or severe neurological manifestations in adults [40]. This led the World Health Organization (WHO) to declare a Public Health Emergency of International Concern (PHEIC) in February 2016 [41]. JEV is a notable cause of encephalitis in Asia [36]. Although most JEV infected people present only subclinical manifestations, a third of symptomatic cases are fatal and almost 50% of survivors present long-term neurological sequelae [42]. WNV is the worldwide most distributed mosquito-borne flavivirus [27]. Infection is mainly asymptomatic in humans, and when symptoms appear, they can range from a mild febrile disease and non-specific flu-like symptoms to a severe neuroinvasive disease that can also lead to a fatal outcome [27]. YFV is currently endemic in over 40 countries in Africa and the Americas. Individuals infected with YFV can present with a wide spectrum of symptoms, ranging from asymptomatic to severe illness with bleeding, jaundice, and death, and, despite vaccination campaigns, over 30,000 deaths are reported each year [43]. 

The *Hepacivirus* genus main representative is the hepatitis C virus (HCV), a major human pathogen that causes liver disease with high risk of developing life-threatening complications, such as liver cirrhosis and hepatocellular carcinoma [44]. Its discoverers, Harvey J. Alter, Charles Rice, and Michael Houghton, were recognized with the 2020 Nobel Prize in Medicine [45]. HCV is mainly transmitted by the parenteral route, although sexual transmission has also been reported [46]. 

The *Pestivirus* genus includes economically important members such as bovine viral diarrhea virus (BVDV) and classical swine fever virus (CSFV) [47]. 

*Pegivirus* genus shows distant sequence similarity to other members of the family, and infections have not been clearly associated with disease, except for non-Hodgkin’s lymphoma [48].

*Flaviviridae* viral replication is initiated by entering in host cells via receptor-mediated endocytosis. Infection is triggered by binding of virions to their cellular receptor, fusion of the viral envelope with the endosomal host membrane, and subsequent release of the viral genome into the cytosol, which is dependent upon the acidic environment within the lysosome [27,49]. A common feature of *Flaviviridae* replication is the formation of virus-induced remodeled membrane organelles. They take advantage of host lipids and proteins to generate these virus-induced membrane compartments to assist in replication [50]. Finally, the viral genome is translated into a single polyprotein and processed to produce mature viral proteins, which are transported through the host endoplasmic reticulum (ER)-Golgi secretory pathway to the cell surface for viral release from infected cells (Figure 2).

Flaviviruses present a worldwide threat to human and animal health, and have the potential to emerge and outbreak in non-endemic geographical regions [27,51], as occurred in the recent Zika virus pandemic [29]. Effective vaccines for humans or animals are only available for a subset of family members, such as tick-borne encephalitis virus (TBEV) YFV, DENV, JEV and, for equids, WNV.

In the quest for antivirals, two different approaches are being applied: searching for compounds directed to viral targets (direct-acting antivirals, (DAAs)), or to cellular components necessary for the viral life cycle (host-directed antivirals (HDAs)) [52,53,54,55]. In the case of the hepacivirus, HCV, the recent development of highly effective DAAs that cure most infections has represented an outstanding success [56]; however, no specific therapies are available for pestiviruses, pegiviruses, or flaviviruses. In the latter case, palliative treatments focused on alleviating patient symptoms rather than combating the virus are in use [57]. Hence, successful identification of antiviral candidates is considered one of the milestones in the fight against this group of pathogens.

Treatment with DAAs often fails due to the rapid generation of drug-resistant viruses [58], as exemplified by resistance-associated substitutions (RASs, amino acid substitutions in the viral proteins) in the case of HCV infections [59,60], thus emphasizing the clinical challenge these resistance mutations represent for the management of patients with HCV and other viral infections [61]. Since almost all members of the *Flaviviridae* family share common host factors, HDA-based therapies are a feasible solution to overcome these drawbacks [41]. Moreover, targeting host proteins required by different viral species can provide a broad-spectrum therapy. However, HDAs show higher toxicity [62] and smaller efficacy window than DAAs [63].

Studies of the flavivirus replication cycle and interaction with the host cell have provided important understanding of essential aspects of their molecular and cellular biology. The most cost and time effective strategy for the development of broad-spectrum antivirals is drug repurposing. Consequently, pharmacological compounds that target host functions key to the viral life cycle are being tested for activity against multiple viruses [41,64,65,66,67,68]. Chemicals targeting the cellular kinome have been explored as novel potential targets for antiviral drug development. In vitro inhibition of flavivirus and hepacivirus infections has been reported upon the administration of different PK-targeting molecules, both activators and inhibitors, thus indicating their potential therapeutic value in the control of these pathogens. The mechanisms by which members of the *Flaviviridae* family interact with PKs are diverse. Several cellular pathways have been identified as potentially involved in flaviviral infections. For instance, Src-family kinase (SFK) inhibitors were reported to block DENV infection by altering virus assembly and secretion [69], and modulation of CAMKII activity impacted attachment of JEV to the host cell surface and viral entry [70]. PKA activity affected ZIKV replication at the post-entry stage by affecting negative-sense RNA synthesis, and HCV infection induces PKA activation to enhance virus entry and infectivity [71].

## 2. Protein Kinase Targets in the Control of Virus of the *Flaviviridae* Family

Pharmacological modulation (both inhibition and activation) of multiple host PKs has been shown to be involved in the regulation of viral infection. Those host PKs most relevant to the control of infection by viruses of the *Flaviviridae* family are listed in Table 4.

*Flaviviridae* family infections can be enhanced or diminished upon treatment with PK pharmacological modulators. The effect of drugs regulating PKs, both activators and inhibitors, in infections by members of this family is further described according to the phylogenetic classification of Manning et al. [11].

### 2.1. The AGC Kinase

AGC kinase group is named for the initials of its members, kinases related to cAMP-dependent protein kinase 1 (PKA), cGMP-dependent protein kinase (PKG), and protein kinase C (PKC). The group is formed by serine/threonine protein kinases that share common characteristic structural features, including the presence of a hydrophobic sequence motif close to the C-terminal lobe of the catalytic core [110]. The group comprises more than 60 members classified into 14 subfamilies: PDK1, AKT/PKB, SGK, PKA, PKG, PKC, PKN/PRK, RSK, NDR, MAST, YANK, DMPK, GRK, and SGK494.

Drugs targeting AGC kinases have been shown to be valuable pharmacological candidates for targeting distinct flaviviruses. The PKA inhibitor PKI significantly reduces ZIKV replication by inhibiting the synthesis of viral genomes, producing minimal cytotoxicity on human endothelial cells and astrocytes, highly susceptible to ZIKV infection [71]. The PKG inhibitor Rp-8-pCPT-cGMPS drastically decreases DENV replication in human HEK293T cell culture, while the PKG activator 8-Br-PET-cGMP produces an increase in DENV yield [20]. Similarly, WNV has been reported to upregulate PKCs during infection [91], and the PKC inhibitors calphostin C and chelerythrine have been reported to reduce WNV multiplication [92]. In contrast, in vitro number of DENV viral copies increased upon treatment with the PKC inhibitor bisindolylmaleimide I, whilst the opposite effect was observed in baby hamster kidney (BHK-21) cells treated with the PKC activator phorbol 12-myristate 13-acetate, thus indicating that inhibition of PKC activity promotes DENV replication [72].

### 2.2. Calcium Calmodulin Dependent Kinases (CAMK)

The CAMKs are serine/threonine kinases activated by increases in the concentration of intracellular calcium ions (Ca^2+^). The activity of this group is mainly regulated by the Ca^2+^ receptor protein calmodulin (CaM). They are classified into two different types: substrate-specific and multi-functional CAMKs. The former can phosphorylate only a specific substrate, while the latter can phosphorylate multiple targets.

A broad antiviral activity against members of the *Flaviviridae* family has been shown by drugs targeting CAMKs. For instance, SFV785 has selective effects on MAPKAPK5 kinase activity, and has been reported to inhibit DENV and YFV viral yield by altering the co-localization of the structural E protein with the DENV replication complexes. This effect on MAPKAPK5 kinase activity did not inhibit DENV RNA synthesis or translation. [73]. Similarly, inhibition of CHK2 with CHK2 inhibitor II effectively reduced JEV production in a range of human cell lines, such as A549, HEK293T, U87 and BE(2)C [97]. Fluvastatine, an inhibitor of DCLK1, downregulated HCV replication in GS5 cell culture, derived from human hepatoma Huh 7.5 cell line, without exerting any negative effect on cell viability [98]. In addition, silencing of Pim Kinase with siRNA, or pharmacological inhibition with SGI-1776, inhibits HCV at an early entry step when human hepatoma Huh 6 and human primary hepatocyte cell cultures were infected [111].

On the other hand, activation of proteins belonging to the CAMK group has also been reported as being effective. Activation of AMPK with PF-06409577 impaired viral replication in WNV, ZIKV, and DENV infected Vero (monkey) and BHK-21 (hamster) cell lines [74], and other pharmacological activators of AMPK, such as AICAR, metformin, and GSK621 have been described as attenuating ZIKV replication in endothelial cell culture [86]. Likewise, liraglutide, which activates AMPK in an AMPK/TORC2-dependent pathway, inhibits HCV replication in the human hepatoma Huh 7 cell line [99].

### 2.3. Casein Kinase 1 (CK1)

CK1 is a monomeric serine-threonine protein kinase with seven isoforms. Pharmacologic inhibition with d4776 was reported to decrease YFV yield in infected human HEK293 cells [96]; however, inhibition of the CK1ε isoform with IC261 promotes WNV infection by suppressing the production of type I interferon, either in vitro, after infection in human HEK293 cells, or using an in vivo model, since CK1ε-deficient mice produced less IFN-β and were more susceptible to WNV infection [112]. On the other hand, the specific CKII inhibitor, 2-dimethylamino-4,5,6,7-tetrabromo-1H-benzimidazole (DMAT), was shown to disrupt virion biogenesis in human hepatoma Huh 7.5 cell infected with HCV [100]. This inhibitor was described as enhancing HCV genotype 1a production in the same cell line [113], thus revealing that genotype-specific differences should be taken into account for potential future pharmacological use of this compound.

### 2.4. CMGC Kinases

CMGC kinases, such as the AGC group, are named with the initials of family members; cyclin-dependent kinase (CDK), mitogen-activated protein kinase (MAPK), glycogen synthase kinase (GSK), and CDC-like kinase (CLK). This group consists of 63 family members highly conserved in eukaryotic organisms.

Drugs targeting CMGC kinases have been described as antiviral candidates against several flaviviruses, as well as against HCV. In the case of DENV, different studies have highlighted the MAPK/ERK pathway as essential for replication, since DENV infection can directly activate proteins in this pathway, including JNK, p38, NTRK1, MAPKAPK5, and c-src/FYN kinases [84]. JNK and p38 kinase inhibitors were reported to significantly reduce DENV protein synthesis and viral yield in infected monocyte-derived macrophages obtained from human peripheral blood [75]. The p38 inhibitor SB203580 prevented lymphopenia, hematocrit increase, and inflammation in human PBMCs, THP-1, and KU812 cell lines infected with DENV [76], and improved the survival rate in DENV-infected AG129 mice [77]. In vitro inhibition of ZIKV virion production was also observed with this agent in infected human endothelial cells and astrocytes [71] and with the related SB202190 [87]. Furthermore, ZIKV production in human neural cell lines was hindered upon treatment with structurally unrelated CDK inhibitors, such as seliciclib, PHA-690509 [114], and Cdk1/2 inhibitor III [78], which also suppressed DENV and JEV viral propagation in the human hepatoma Huh 7 cell line. Selective inhibition of the MAPK/ERK pathway has also been described to block infectious HCV production in infected human Huh 7.5 cells [101], and the inhibitor BmKDfsin3, obtained from scorpion (*Mesobuthus martensii*) venom, also decreases HCV replication by downregulation of the p38 MAPK signal pathway in Huh7.5.1 and HEK293T infected cell lines [102]. Finally, an SRPK inhibitor (SRPIN340) suppressed the expression of an HCV subgenomic replicon and the in vitro replication in Huh7 and Huh7.5.1 cell lines of the HCV-JFH1 clone in a dose-dependent manner [103].

### 2.5. Tyrosine Kinases (TKs)

TK phosphorylates almost exclusively on tyrosine residues, whilst most other kinases are selective for serine or threonine. This group is classified into two subtypes, receptor (RTKs) and non-receptor, or cytoplasmic TKs (CTKs), depending on their function in transmembrane signaling, or within the cell mediating signal transduction to the nucleus, respectively. RTKs have transmembrane and extracellular domains, whilst CTKs do not. RTKs primarily transmit extracellular signals into the cell. CTKs are, generally located within the cytoplasm, although often membrane-associated.

TKs have been deeply studied and their involvement in flavivirus replication has been widely reported [73,83,88,93,104]. The c-Src/Fyn kinase has been identified as a cellular target in DENV RNA replication. The pharmacological inhibitor saracatinib (AZD0530) inhibits virion assembly of DENV in human Huh7 and HEK293T infected cell lines [69,79]. Likewise, compound 16i, another Src inhibitor, was reported to suppress DENV replication at low micromolar concentrations with no significant toxicity to the host cell [81], thus validating the Src family of TKs as potential drug targets for the development of treatments against DENV infection. Other SFKs are also implicated in DENV infection; Abl inhibitor GNF-2 interferes with DENV replication in human hepatoma Huh-7 and Vero African green monkey kidney infected cells [80]. The involvement of other TKs, such as those acting on the JAK/STAT3 pathway, has been described. JAK2 and JAK3 inhibitors have been reported to reduce DENV-induced phosphorylation of STAT3 and cell migration, as well as production of the chemokines IL-8 and RANTES in infected hepatocytes [82]. Likewise, WNV-infected human SK-N-MC and HEK 293 cells treated with the SFK inhibitor PP2 show a decrease in viral titers, whilst there was no effect on intracellular levels of either viral RNA or protein, thus suggesting that the drug has no effect on the early stages of replication [93]. Two inhibitors of AXL phosphorylation, cabozantinib, and R428, significantly impair ZIKV infection of human endothelial cells [88]. TKs have also been related to hepacivirus replication, and the EGFR inhibitor erlotinib inhibited HCV infection in a dose-dependent manner in different cell lines, such as Huh7, Huh7.5.1 cells and primary human hepatocytes [104].

Additionally, diverse TKs have been described as broad-spectrum anti-viral agents. A covalent host BTK inhibitor, QL-XII-47, was reported to inhibit DENV, WNV, and ZIKV in the human Huh 7 cell line [83]. Furthermore, the kinase inhibitor SFV785 was shown to reduce secretion of infectious DENV and YFV virions in Vero and BHK-21 infected cells [73]. Likewise, inhibition of EGFR kinase activity via induction of IFN-α inducible protein 6 (IFI6), an IFN-stimulated gene (ISG), strongly inhibited DENV either in vitro or in vivo in AG129 mice [115], WNV [94], and HCV infection [116], either in vitro or in vivo, in AG129 mice. The wide spectrum TK inhibitor dasatinib was reported to reduce virion assembly in DENV via Fyn kinase in human Huh7 and HEK293T infected cell lines, and to inhibit HCV infection via EphA2 TK in different cell lines, such as Huh7, Huh7.5.1 cells, and primary human hepatocytes [69,104].

### 2.6. Tyrosine Kinase-Like (TKL)

TKL kinases are serine-threonine protein kinases with sequence similarity to TKs, but lacking TK-specific motifs. This is the most recently defined PK group, and families within it are little related to each other. As with TKs, TKL kinases are classified into receptor and non-receptor kinases, and are distributed in eight major families.

The main target among TKL kinases reported as antiviral candidates against flaviviruses are Receptor Interacting Protein Kinases (RIPKs), key mediators of cellular signaling that are essential for the early control of diverse pathogens [117]. Among them, RIPK3 has been described as involved in neuroinflammation and neuronal death during JEV infection, tested either in vitro using neuro2a cells or in vivo, in wild type and RIPK3–/– mice [118]. RIPK3 signaling also restricted viral replication in ZIKV [89] and WNV [95] infections in mice.

### 2.7. Other PKs

There are several families included in the ePKs identified by Manning [11] that lack sequence similarity with the previously described ePK groups, and, thus, they are catalogued in a separate group.

Numerous drugs targeting this heterogeneous group have shown antiviral activity against flaviviruses and HCV infections. For instance, the IRE1 kinase inhibitor KIRA6 reduces viral RNA levels in ZIKV infected the human HeLa cell line [90]. DENV infection has been widely reported to be inhibited by inhibitors of different members of the group, as exemplified by treatment with pyruvate kinase PKM2 inhibitor in DENV-infected U937 cells [84], the AurKB inhibitor ZM 447439 in DENV-infected Huh-7 cells [85], and the NAK family inhibitors, sunitinib and erlotinib (AAK1 and GAK subfamilies inhibitors respectively) in DENV-infected Huh-7 cells [18]. PKR is modulated by cyclophilin A, triggering antiviral responses to inhibit HCV infection in Huh 7 cell line [119], and the PKR2 inhibitor HA1077, also known as fasudil restricted HCV replication in mice [105]. NAK inhibitors also affect HCV assembly, which was disrupted by treatments with erlotinib, dasatinib, or isothiazolo[5,4-b]pyridine (GAK subfamily inhibitors) in Huh 7.5 cells [106], and sunitinib or PKC-412 (AAK1 subfamily inhibitors) tested in Huh7.5 and 293T cell lines [107,108]. Erlotinib and dasatinib are not NAK specific, and inhibit TKs, as mentioned above, although the authors of these studies pointed to GAK inhibition as the cause of HCV inhibition [106]. BX795, a TBK1/IKKε inhibitor, showed effects against HCV infection in the Huh 7 cells [109]. As a consequence of its involvement in autophagy, endoplasmic reticulum (ER) stress, and unfolded protein response (UPR), PERK has been associated with apoptosis in JEV infection either in vitro in neuro2a and BHK-21 cells or in vivo in BALB/c mice [120], DENV infected canine MDCK cells [121,122], and WNV infected SK-N-MC human neuroblastoma cells [123].

Pharmacological modulation of the STE, RGC, and atypical kinases has yet to be linked to flaviviral infection.

## 3. Conclusions

PKs play central roles in cellular signaling pathways through modulating protein phosphorylation in processes such as cell growth, differentiation, and metabolism. PKs have been deeply studied and characterized and represent attractive targets for drug design against a wide spectrum of disorders, including viral diseases, where, to date, they are the major host target for antiviral pharmacological development. Host cell kinases are involved in every step of the viral life cycle in a wide range of viral species, including those of the *Flaviviridae* family. The identification of antiviral candidates for flaviviral epidemics is considered one of the milestones in the fight against these health-threatening pathogens. Two different approaches are being applied in the quest for antivirals: DAAs, directly aimed to viral targets, or HDAs that target cellular components essential for the viral life cycle. DAAs often fail due to the rapid emergence of drug-resistant viruses, making HDA-based antivirals a reasonable solution to overcome these drawbacks. Moreover, targeting host proteins required by different viral species may provide a broad-spectrum therapy and, thus, the use of chemicals targeting the cellular kinome has been explored in the search for new antivirals. Nevertheless, and even though there is a plethora of promising studies concerning the inhibition of flavivirus and HCV infection through the use of different PKs-targeting cellular molecules, which could lead to broad-spectrum repurposed or new antivirals, none has yet been approved. Therefore, there is still a long way to go in the study of PK modulators as therapeutic measures to combat these animal and human pathogens.

## Figures and Tables

**Figure 1 ijms-21-09524-f001:**
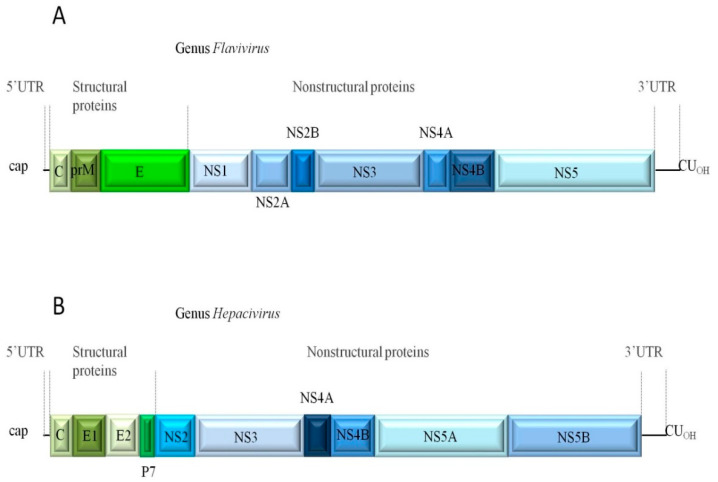
Schematic view of the genomic organization of flaviviruses (**A**) and hepaciviruses (**B**). UTR: untranslated region; C: capsid or core protein; prM: pre-membrane protein; p7: polypeptide 7; E: envelope protein; NS: non-structural proteins.

**Figure 2 ijms-21-09524-f002:**
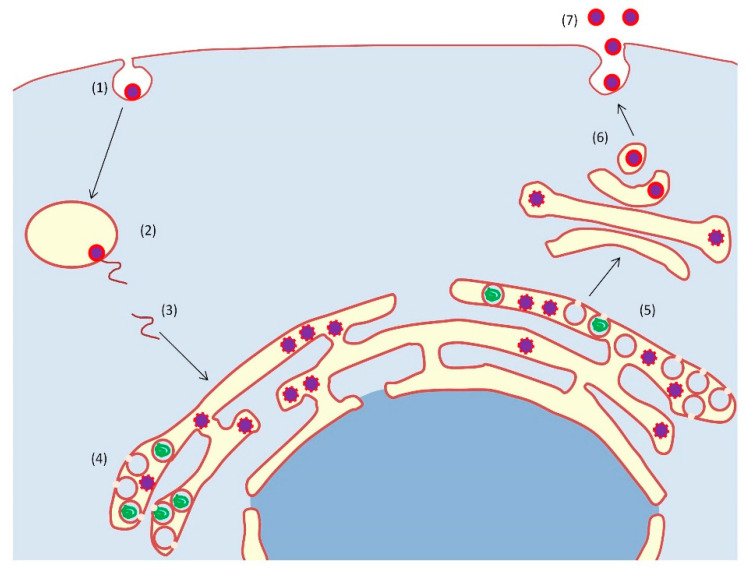
Schematic representation of flaviviral infection. Viral particles enter the cell via receptor-mediated endocytosis (1). The acid environment of endosomes allows viral and endosomal membranes fusion (2), and the subsequent release of viral RNA into the cytosol (3), supporting genome replication and particle biogenesis (4). Viral RNA is translated and processed to produce viral proteins (5), which travel through the Golgi apparatus (6), allowing particle release (7).

**Table 1 ijms-21-09524-t001:** Classification of eukaryotic protein kinases (ePKs).

Group	Representative Families of the Group
AGC	PKA (cAMP-dependent protein kinase), PKC (protein kinase C), PKG (cGMP-dependent protein kinase), PKN (protein kinase N), AKT (protein kinase B)
CAMK (Calcium Calmodulin dependent kinase)	PhK (phosphorylase kinase), CAMK (Ca2+/calmodulin-dependent protein kinase), MAPKAPK (mitogen-activated protein kinase-activated protein kinase), MLCK (myosin light-chain kinase)
CK1 (Casein Kinase 1)	TTBK (tau-tubulin kinase), VRK (vaccinia-related kinase)
CMGC	CDK (cyclin-dependent kinase), MAPK (mitogen-activated protein kinase), GSK (glycogen synthase kinase), CDKL (cyclin Dependent Kinase Like), JNK (c-Jun N-terminal kinase), p38
STE	STE7 (sterile 7), STE11 (sterile 11), STE20 (sterile 20)
TK (Tyrosine kinase)	EGFR (epidermal growth factor receptor), PDGFR (platelet-derived growth factor receptors), JAK (Janus kinase), Eph (erythropoietin-producing human hepatocellular receptors), Fyn (proto-oncogene tyrosine-protein kinase), SFK (Src-family kinase), TRK (tropomyosin receptor kinase)
TKL (Tyrosine kinase-like)	MLK (mixed lineage kinase), LISK, IRAK (interleukin-1 receptor-associated kinase), RIPK (receptor-interacting serine/threonine-protein kinase)
RGC (Receptor guanylate cyclase)	RGC (receptor guanylate cyclase)
Others	MYT (membrane-associated tyrosine- and threonine-specific cdc2-inhibitory kinase), ULK (Unc-51 like autophagy activating kinase), PLK (polo-like kinase), SCY, NKF (new kinase family), NAK (numb-associated kinase), PEK (pancreatic eukaryotic initiation factor-2alpha kinase)

**Table 2 ijms-21-09524-t002:** Classification of atypical protein kinases (aPKs).

Group	Representative Families of the Group
alpha	ChaK (Channel kinase), eEF2K (Eukaryotic elongation factor 2 kinase)
PIKK (phosphatidyl inositol 3′ kinase-related kinase)	ATM (Ataxia telangiectasia mutated kinase), ATR (Ataxia telangiectasia and Rad3 related kinase), FRAP, SMG1 (Nonsense Mediated MRNA Decay Associated PI3K Related Kinase)
PDHK (pyruvate dehydrogenase kinase)	PHDK (pyruvate dehydrogenase kinase), BKCDK
RIO (right open reading frame)	RIOK (Right open reading frame protein kinase), SUDD (Right open reading frame protein kinase3)

**Table 3 ijms-21-09524-t003:** Members of the *Flaviviridae* family. The four genera, with representative members and genome size corresponding to each genus, are displayed.

Genus	Representative Members *	Genome Size (Kb)
*Flavivirus*	YFV, WNV, DENV, ZIKV, TBEV	9.2–11
*Hepacivirus*	HCV	8.9–10.5
*Pestivirus*	BVDV, CSFV	12.3–13
*Pegivirus*	GBV-A, HPgV	8.9–11.3

* YFV: yellow fever virus; WNV: West Nile virus; DENV: dengue virus; ZIKV: Zika virus; TBEV: tick borne encephalitis virus; HCV: hepatitis C virus; BVDV: bovine viral diarrhea virus; CSFV: classical swine fever virus; GBV-A: GB virus A; HPgV: human pegivirus type 2.

**Table 4 ijms-21-09524-t004:** Members of the *Flaviviridae* family inhibited by PK regulation.

Virus	PK	Kinase Group	Kind of PK Regulation	Inhibitor/Activator Used	References
	PKG	AGC	Inhibition	Rp-8-pCPT-cGMPS, TEA	[20]
	PKC	AGC	Activation	Phorbol 12-myristate 13-acetate (PMA)	[72]
	MAPKAPK5	CAMK	Inhibition	SFV785	[73]
	AMPK	CAMK	Activation	PF-06409577	[74]
	JNK	CMGC	Inhibition	SP60025	[75]
	P38	CMGC	Inhibition	SB003580	[75,76,77]
	CDK	CMGC	Inhibition	Alsterpaullone 2-cyanoethyl, Cdk1/2 inh III,Cdk2/9 inh	[78]
DENV	SFK	TK	Inhibition	AZD0530, Dasatinib, GNF-2	[69,79,80,81]
	JAK	TK	Inhibition	WHI-P131	[82]
	BTK	TK	Inhibition	QL-XII-47	[83]
	NTRK1	TK	Inhibition	SFV785	[73]
	PKM2	Other	Inhibition	PKM2 inhibitor	[84]
	AurKB	Other	Inhibition	ZM 447439	[85]
	NAK	Other	Inhibition	Sunitinib, Erlotinib	[18]
	PKA	AGC	Inhibition	PKI 14-22	[71]
	AMPK	CAMK	Activation	PF-06409577 AICAR, Metformin, GSK621	[74,86]
	P38	CMGC	Inhibition	SB203580, SB202190	[71,87]
ZIKV	AXL	TK	Inhibition	Cabozantinib, R428	[88]
	BTK	TK	Inhibition	QL-XII-47	[83]
	RIPKs	TKL	Activation	AP1	[89]
	IRE1 K	Other	Inhibition	KIRA 6	[90]
	PKC	AGC	Inhibition	Calphostin C, Chelerythrine	[91,92]
	AMPK	CAMK	Activation	PF-06409577	[74]
	SFK	TK	Inhibition	PP2	[93]
WNV	BTK	TK	Inhibition	QL-XII-47	[83]
	EGFR	TK	Inhibition	IFN-α inducible protein 6	[94]
	RIPKs	TKL	Activation	AP1	[95]
	MAPKAPK5	CAMK	Inhibition	SFV785	[73]
YFV	CK1	CK1	Inhibition	D4776	[96]
	NTRK1	TK	Inhibition	SFV785	[73]
JEV	CHK2	CAMK	Inhibition	CHK2 inhibitor II	[97]
	CDK	CMGC	Inhibition	Alsterpaullone 2-cyanoethyl, Cdk1/2 inh III,Cdk2/9 inh	[78]
	DCLK1	CAMK	Inhibition	Fluvastatine	[98]
	AMPK	CAMK	Activation	Liraglutide	[99]
	CKII	CK1	Inhibition	2-dimethylamino-4,5,6,7-tetrabromo-1H-benzimidazole	[100]
	MAPK/ERK	CMGC	Inhibition	PD98059, Sorafenib	[101]
HCV	P38/MAPK	CMGC	Inhibition	BmKDfsin3	[102]
	SRPK	CMGC	Inhibition	SRPIN340	[103]
	EGFR	TK	Inhibition	Erlotinib, Dasatinib	[69,104]
	PKR	Other	Inhibition	HA1077	[105]
	NAK	Other	Inhibition	Isothiazolo [5,4-b]pyridine, Sunitinib, PKC-412	[106,107,108]
	TBK1/IKKε	Other	Inhibition	BX795	[109]

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
