# Peer review of "Potential for Protein Kinase Pharmacological Regulation in Flaviviridae Infections"

_ijms, 2020, doi:10.3390/ijms21249524_

Round 1

Reviewer 1 Report

The manuscript “Protein Kinase Pharmacological Regulation in Flaviviridae Infections” is not having a definite conclusion. Although the title of the manuscript describes “Flaviviridae Infections”, but there are no proper description of Flaviviridae viruses. It has not been mentioned how these viruses cause infection. The authors have not mentioned the mode of action of Flaviviridae viruses-induced infections. Host is also not clear, where it is human, animal, bird, or some other organism.

Authors have focused Protein kinases and their inhibitors. But the details given are superficial. The cited articles have not been elaborated properly. For example:

“For instance, the PKA inhibitor PKI significantly reduces ZIKV virion production without exerting toxicity on the host cells [60]”

The above sentence is not properly covering the theme of reference number [60]. It does not show what is meant by toxicity mentioned in reference number [60]. Additionally, it is not making clear to the reader what the “host cells” are.

Similarly, regarding “STE kinase group” authors have added the following paragraph:

“The name STE stands for Sterile, since the group includes homologs of the yeast sterile kinases, and consists of 3 main families (STE7, STE11 and STE20). The STE20 family is the largest one and is divided into many subfamilies. Some of them are implicated in MAPK cascades by sequentially activating one to each other to finally activate the MAPK family (belonging to CMGC PKs group), while other members of this group may have completely different functions. Currently, no chemical regulation of this PKs group has been associated to infections with any member of the Flaviviridae family”.

In the above paragraph, the authors describe that “Currently, no chemical regulation of this PKs group has been associated to infections with any member of the Flaviviridae family”. So what is the significance of including “STE kinase group” in this manuscript?

Likewise, regarding the “Receptor guanylate cyclase (RGC) kinase group”, authors have mentioned that there is no connection of this group with the flaviviral infections.

After critical review, and looking at the reputation and high impact of “International Journal of Molecular Sciences”, I do not recommend this manuscript for publication.

Author Response

The manuscript “Protein Kinase Pharmacological Regulation in Flaviviridae Infections” is not having a definite conclusion. Although the title of the manuscript describes “Flaviviridae Infections”, but there are no proper description of Flaviviridae viruses. It has not been mentioned how these viruses cause infection. The authors have not mentioned the mode of action of Flaviviridae viruses-induced infections. Host is also not clear, where it is human, animal, bird, or some other organism.

Following reviewer´s comments, Flaviviridae family has been better described including animal hosts and  infections in humans (page 4 first paragraph). Likewise, a new paragraph has been included with detailed information about virus replication (page 5)

Authors have focused Protein kinases and their inhibitors. But the details given are superficial. The cited articles have not been elaborated properly. For example:

“For instance, the PKA inhibitor PKI significantly reduces ZIKV virion production without exerting toxicity on the host cells [60]”

The above sentence is not properly covering the theme of reference number [60]. It does not show what is meant by toxicity mentioned in reference number [60]. Additionally, it is not making clear to the reader what the “host cells” are.

As suggested, this sentence has been changed for a better covering of the theme of the reference (Suppression of Zika Virus Infection and Replication in Endothelial Cells and Astrocytes by PKA Inhibitor PKI 14-22. J. Virol. 2018 PMID 29212931)

Similarly, regarding “STE kinase group” authors have added the following paragraph:

“The name STE stands for Sterile, since the group includes homologs of the yeast sterile kinases, and consists of 3 main families (STE7, STE11 and STE20). The STE20 family is the largest one and is divided into many subfamilies. Some of them are implicated in MAPK cascades by sequentially activating one to each other to finally activate the MAPK family (belonging to CMGC PKs group), while other members of this group may have completely different functions. Currently, no chemical regulation of this PKs group has been associated to infections with any member of the Flaviviridae family”.

In the above paragraph, the authors describe that “Currently, no chemical regulation of this PKs group has been associated to infections with any member of the Flaviviridae family”. So what is the significance of including “STE kinase group” in this manuscript?

Likewise, regarding the “Receptor guanylate cyclase (RGC) kinase group”, authors have mentioned that there is no connection of this group with the flaviviral infections.

As the manuscript summarizes the interactions between the different groups of protein kinases described and viruses of the Flaviviridae family, we think that, although the STE and TGC kinase groups have not yet been related to them, they should be mentioned to highlight it and avoid misinterpretation.

After critical review, and looking at the reputation and high impact of “International Journal of Molecular Sciences”, I do not recommend this manuscript for publication.

Contrary to reviewer´s opinion, and in agreement with reviewer 2, we believe that, after improving the manuscript following both reviewer´s comments, the manuscript indeed is of interest for the International Journal of Molecular Sciences´s readers, and, thus, we hope the Editorial board find it now suitable for its publication

Reviewer 2 Report

In this review article, the authors described the possible protein kinase pharmacological regulation in Flaviviridae infections. The overall quality is good but, in my opinion, is necessary to better describe the functional mechanisms explaining the link between PKs and Flaviviridae infections. i.e. authors stated, "the PKA inhibitor PKI significantly reduces ZIKV virion production without exerting toxicity on the host cells". Why? Which are the molecular mechanisms/processes modulated after PKA inhibition that can explain for this physiological effect? Please clarify in detail these functional aspects.

Minor

Table 1 and 2. I suggest adding a line reporting protein names abbreviations.

Author Response

In this review article, the authors described the possible protein kinase pharmacological regulation in Flaviviridae infections. The overall quality is good but, in my opinion, is necessary to better describe the functional mechanisms explaining the link between PKs and Flaviviridae infections. i.e. authors stated, "the PKA inhibitor PKI significantly reduces ZIKV virion production without exerting toxicity on the host cells". Why? Which are the molecular mechanisms/processes modulated after PKA inhibition that can explain for this physiological effect? Please clarify in detail these functional aspects.

We thank the reviewer for her/his favorable opinion about the interest of our study and her/his comments to improve it.

Following reviewer´s suggestions, this sentence has been changed to better explain the mechanisms by which ZIKV is inhibited after PKA modulation

Minor

Table 1 and 2. I suggest adding a line reporting protein names abbreviations.

As requested for the reviewer, protein names have been reported in tables 1 and 2.

Round 2

Reviewer 1 Report

All the suggested changes were incorporated in the review. The manuscript is now acceptable for publication. However, authors should remove the following sections from the manuscript:

STE kinase group, and

Receptor guanylate cyclase (RGC) kinase group

Author Response

Thanks to the reviewer for her/his comments and decision. Sections of STE kinase and Receptor guanylate cyclase groups have been removed from the manuscript

Reviewer 2 Report

The manuscript has been improved and it is now suitable for the publication

Author Response

We are very grateful with reviewer´s decision.